# Effects of Stocking Density, Size, and External Stress on Growth and Welfare of African Catfish (*Clarias gariepinus* Burchell, 1822) in a Commercial RAS

Björn Baßmann [1,*], Lisa Hahn [1,2], Alexander Rebl [3], Lisa Carolina Wenzel [1], Marc-Christopher Hildebrand [1], Marieke Verleih [3] and Harry Wilhelm Palm [1]

1 Department of Aquaculture and Sea-Ranching, Faculty of Agricultural and Environmental Sciences, University of Rostock, Justus-von-Liebig-Weg 6, 18059 Rostock, Germany
2 Department of Marine Biology, Institute for Biological Sciences, University of Rostock, Albert-Einstein Straße 3, 18055 Rostock, Germany
3 Fish Genetics Unit, Institute of Genome Biology, Research Institute for Farm Animal Biology (FBN), Wilhelm-Stahl-Allee 2, 18196 Dummerstorf, Germany
* Correspondence: bjoern.bassmann@uni-rostock.de; Tel.: +49-(0)381-498-3745

**Abstract:** The effects of semi-intensive (100 kg m$^{-3}$), intensive (200 kg m$^{-3}$), and super-intensive (400 kg m$^{-3}$) stocking densities on the growth and welfare of African catfish (*Clarias gariepinus*) were investigated under commercial production conditions. Plasma cortisol, glucose, and selected transcripts following a stress challenge, lactate, as well as skin lesions, were analyzed at regular intervals (from 12 g juveniles to 1.5–2.0 kg). The fish grew well, but after 23 weeks, the semi-intensively stocked fish had a mean final weight of 1830.5 g, significantly higher than the super-intensively stocked fish with 1615.4 g, and considerably higher than the intensively stocked fish with 1664.8 g ($p > 0.05$). Cortisol and glucose responses significantly differed between stressed and unstressed fish, but not between treatment groups. An unforeseen external stressor (nearby demolition noise) caused stress responses among all treatment groups, but was similarly coped with. Mortality ranged between 3.8–9.2%. In the juveniles, skin lesions were reduced under intensive or super-intensive densities, with the least under semi-intensive densities in outgrown fish. Expression profiles of 22 genes were compared in the spleen at semi-intensive and super-intensive densities. The transcript concentrations of most genes remained unchanged, except for *slc39a8* and *mtf1*, which were significantly downregulated in stressed catfish under semi-intensive conditions. We demonstrated that African catfish growth performance and welfare depend on age and stocking density, also reacting to demolition noise. This supports farm management to optimize stocking densities during the grow-out of African catfish in RAS and suggests avoiding external stress.

**Keywords:** cortisol; demolition noise; fish well-being; grading; growth performance; recirculating aquaculture; mortality

## 1. Introduction

In finfish aquaculture, stocking density is a pivotal factor and an issue of frequent debate, as it may be a source of chronic stress, leading to physiological alterations, including stress responses, growth reduction, and impairment of health [1–3]. Based on these indicators, fish welfare can be considered diminished [2,4–7]. According to Ellis et al. [2] (p. 494), "… the term 'stocking density' refers to the concentration at which fish are initially stocked into a system" (*sensu strictu*). However, most often, the term is used to describe the density of fish at any time. It may thus be understood as a dynamic factor, since the actual density increases or decreases as the fish grow or are removed from the rearing volume. Fish species in aquaculture are stocked at very different densities, typically ranging from < 10 to 100 kg m$^{-3}$ [4,8]. This widely varies due to the different needs and/or tolerances of

the respective species. Similarly, the impact of a specific stocking density on the species' welfare may differ as well [6]. In Europe, fish farming is often industrialized and highly intensive [9]. According to Naylor et al. [10] (p. 1017), "intensification implies increasing the density of individuals, which requires greater use and management of inputs, greater generation of waste products, and increased potential for the spread of pathogens". In this respect, particularly high, but also low stocking densities, may result in stress, poor fish health, and welfare [11–13], subsequently having a negative impact on the fish performance, which in turn may also have economic consequences [14,15].

A species whose global production in aquaculture has distinctly increased in recent years is the African catfish (*Clarias gariepinus* Burchell, 1822). In Germany, production in recirculating aquaculture systems (RASs) increased from 318,575 kg per year$^{-1}$ to 1,193,137 kg per year$^{-1}$ between 2011 and 2019 [16,17]. In moderate climates, this warm-water species is farmed in RASs, while ponds are predominantly used in tropical and subtropical regions [18]. The African catfish has a very good feed conversion ratio (FCR), a high growth potential, and is highly tolerant towards adverse environmental conditions, such as low oxygen or elevated nitrogen compounds in the water [19,20]. This is based on its biological adaptations to its natural habitats, which are often characterized by shrinking water bodies or complete drainage during the dry season. Especially favorable is their ability to utilize atmospheric oxygen with their arborescent organs [21]. For this reason, African catfish are adapted to survive for some time, densely packed inside smaller pools or even in moist substrates and mud [20]. This attribute is utilized in commercial aquaculture by stocking this species up to super-intensive densities (max. 500 kg m$^{-3}$) [22].

Reduced growth, alteration of physiological processes, increased stress responses, and/or rise in mortality were described under high stocking densities [1,23–27]. According to van de Nieuwegiessen et al. [28], final densities between approx. 100–500 kg m$^{-3}$ did not affect the growth and welfare of African catfish in a size range from approx. 1.0–1.5 kg fish$^{-1}$. In contrast, wellbeing improved with increasing stocking density in fish between approx. 100–300 g at final densities between 16.7–315 kg m$^{-3}$, as significantly fewer skin lesions occurred at the highest density. According to van de Nieuwegiessen et al. [22], African catfish juveniles between approx. 10–100 g showed a comparable growth performance and mortality at all the tested densities, but there was evidence that growth may be best at medium densities (175 kg m$^{-3}$) and worst at the higher densities (300 kg m$^{-3}$). Plasma cortisol, glucose, and lactate remained unaffected by stocking densities, and fish from both low (50 kg m$^{-3}$) and high densities (300 kg m$^{-3}$) showed no cortisol response to an additional acute stress challenge. The authors suggested an impaired cortisol response related to the downregulation of components of the hypothalamic-pituitary-interrenal (HPI) axis in fish as a result of chronic stress: the adrenocorticotropic hormone (ACTH) or cortisol receptors [22,29]. African catfish fry between approx. 0.1–0.3 g showed no significant differences in growth performance at stocking densities between 10–30 fish L$^{-1}$, but aggressive behavior was less frequent at higher stocking densities [30]. In juvenile African catfish (approx. 10–100 g) under extensive stocking densities (500 and 1125 fish m$^{-3}$), increased agonistic interactions during the first weeks were described. In contrast, mature African catfish from higher stocking densities showed the highest number of skin lesions after stress induction. In general, however, aggression and the number of lesions were described to decrease with age. It was suggested that both low and high stocking densities might impair welfare [22,28].

So far, most findings have originated from experiments in aquaria and not under commercial conditions. Therefore, the present study presents the growth performance and welfare of African catfish throughout an entire grow-out period in a commercial RAS under intensive conditions. We evaluate the dependence of current fish welfare indicators and the expression of several genes on fish size under three different stocking densities. In addition, the effect of an unforeseen external stressor during the experiment is discussed.

## 2. Materials and Methods

### 2.1. Production System, Maintenance, and Water Quality

A commercially scaled RAS at the aquaculture research facility 'FishGlassHouse' of the University of Rostock (Germany) was used for this experiment. It consisted of nine identical rearing tanks (each measuring (L × W × H) $1.8 \times 1.0 \times 0.7$ m, $1.26$ m$^3$), a settling tank ($1.76$ m$^3$, total effective surface area: $108.08$ m$^2$) to remove suspended solids from the water, a trickling filter ($5.6$ m$^2$, specific surface area: $125.00$ m$^2$ m$^{-3}$, total volume: $11.80$ m$^3$, total specific surface area: $1474.20$ m$^2$) for biological water treatment, and a sump ($4.41$ m$^3$). The total system contained approximately $16.90$ m$^3$ of water [31]. An automatic temperature control/heater and a float switch for water level regulation were located in the sump. Tap water was used to replace evaporated water.

The water parameters of temperature, pH, oxygen concentration and saturation, electric conductivity (EC), salinity, and redox potential were controlled each day, after trickling filtration (before the rearing tanks) with a portable multimeter (Hach-Lange HQ40D, Germany). Weekly water samples were taken and analyzed in triplicate with an automatic photometric-analyzer (Gallery™, Thermo Fisher Scientific) to monitor concentrations of ammonium/ammonia ($NH_4^+/NH_3$), nitrite ($NO_2^-$), and nitrate ($NO_3^-$). If the pH dropped below a threshold of 5, lime hydrate was added to the RAS or the water was changed in order to accordingly adjust the water quality. After starting the experiment, the settling tank was cleaned once a week; later, it was cleaned much more frequently to ensure system stability. The water quality parameters in the course of the experiment are summarized in Table 1.

**Table 1.** Water parameters as mean $\pm$ standard deviation (SD) and minimum/maximum values between samplings (T0–T1, T1–T2, etc.). The parameters were measured every day (or weekly *).

|  | Mean $\pm$ SD | Min/Max (T0–T1) | Min/Max (T1–T2) | Min/Max (T2–T3) | Min/Max (T3–T4) | Min/Max (T4–T5) |
|---|---|---|---|---|---|---|
| Temperature [°C] | $26.1 \pm 1.6$ | 24.2/26.1 | 23.8/26.7 | 19.0/27.5 | 23.6/28.0 | 25.7/28.6 |
| $O_2$ [mg L$^{-1}$] | $6.0 \pm 0.9$ | 7.1/8.1 | 6.1/8.3 | 4.1/8.7 | 4.7/6.8 | 4.4/6.2 |
| $O_2$ [%] | $74.0 \pm 10.9$ | 90.4/96.5 | 74.4/103.5 | 50.8/97.1 | 60.1/82.6 | 53.1/80.2 |
| pH | $6.7 \pm 0.9$ | 7.7/8.2 | 3.7/8.1 | 6.1/7.7 | 6.1/7.8 | 5.0/7.2 |
| EC [$\mu$S cm$^{-1}$] | $1382 \pm 451$ | 762/883 | 906/1610 | 672/1520 | 814/2230 | 1371/2480 |
| Salinity [‰] | $0.7 \pm 0.2$ | 0.4/0.4 | 0.4/0.8 | 0.4/0.7 | 0.4/1.1 | 0.7/1.2 |
| RedOx [mV] | $160.1 \pm 31.7$ | 173.6/194.5 | 158.8/276.2 | 120.2/192.0 | 105.7/191.2 | 60.8/174.3 |
| $NH_4$ * [mg L$^{-1}$] | $13.60 \pm 3.88$ | 9.62/18.99 | 9.03/14.73 | 8.80/13.41 | 8.48/16.50 | 8.78/13.21 |
| $NO_2$ * [mg L$^{-1}$] | $0.19 \pm 0.17$ | 0.03/0.03 | 0.09/0.49 | 0.06/0.36 | 0.05/0.35 | 0.07/0.37 |
| $NO_3$ * [mg L$^{-1}$] | $475 \pm 154$ | 591/606 | 511/518 | 339/346 | 325/750 | 344/349 |

### 2.2. Fish Stocking, Feeding, and Growth

Three different treatment groups were compared with a semi-intensive ($100$ kg m$^{-3}$), intensive ($200$ kg m$^{-3}$) and super-intensive ($400$ kg m$^{-3}$) stocking density. On 18 January 2021, a total of 1848 African catfish juveniles of mixed sex were obtained from a local farmer (Fischzucht Abtshagen GmbH & Co. KG, Abtshagen, Germany). The fish were randomly stocked with either 88, 176, or 352 fish in 3 of the 9 tanks (randomized block-design), respectively, to reach an approximate final weight of 100, 200, or 400 kg m$^{-3}$ at a slaughter weight of 1.5 kg. From each stocking density, 100 individual weight and length measurements were recorded as representative samples. Since newly added individuals are often harassed, fish in this study were not graded in order to avoid affecting the respective group compositions, and thus possibly influencing the results.

The fish were fed with a catfish diet (Alltech Coppens, pellet sizes of 1.5–4.5 mm) according to a commercial feeding protocol from Alltech Coppens for African catfish of 10–2000 g at 5.62–0.84% BW d$^{-1}$. Feeding took place every two hours between 07:00 p.m.–05:00 a.m. by using automatic feeders. The FCR, the specific growth rate (SGR), and the condition index (CI) were calculated for each sampling date.

$$FCR = TFI/W_f - W_i \tag{1}$$

with TFI = total feed intake [g], $W_i$ = initial fish weight [g], and $W_f$ = final fish weight [g].

$$SGR = (\ln(W_f) - \ln(W_i))/t \times 100 \tag{2}$$

with $W_i$ = initial fish weight [g] and $W_f$ = final fish weight [g], t = days.

$$CI = \text{fish mass [g]} \times 100/\text{fish length [cm]}^3 \tag{3}$$

After final sampling, the individual weights and total body lengths of 100 fish per stocking density were recorded again. The experiment was expected to last 5 to 6 months, depending on when the fish would reach their slaughter weight.

### 2.3. Sampling

All treatments were carried out in accordance with the EU guidelines 2010/63/EU for animal experiments and were approved by the relevant ethics committee.

The fish of the three treatment groups (semi-intensive, intensive, and super-intensive) were sampled at the beginning (i.e., after stocking and a one-week adaptation period) and regularly afterwards at 6-week intervals throughout the entire growth phase. In each sampling, three fish per tank (nine fish per treatment group) were anesthetized (eugenol bath, dosage: 50 mg $L^{-1}$) and blood sampled over their caudal blood vessels within 5 min. Another three fish per tank (nine fish per treatment group) were anesthetized and blood sampled after additional induced stress; i.e., after netting and 30-min confinement in a tub without water but under humidification. According to Martins et al. [32], air exposure up to 1 h is a rather moderate stressor for African catfish. In each case, blood glucose and lactate levels were determined in situ using test strips (Roche, Accu Check Aviva/Accutrend Plus, Mannheim, Germany). The remaining blood was anticoagulated (BD Vacutainer, 5.4mg K-EDTA, Franklin Lakes, NJ, USA) and stored on ice for later centrifugation (Hettich Universal 320 R, 10 min at 10,000 rpm and 4 °C, Tuttlingen, Germany). The plasma was separated from the cells. Cortisol was analyzed in the plasma by enzyme-linked immunosorbent assay (ELISA, Cusabio fish cortisol, sensitivity: 0.0023 ng $mL^{-1}$, Wuhan, China) according to the manufacturer's instructions using a micro-plate reader at 450 nm (iMark, Bio-Rad, Hercules, CA, USA). The cortisol concentrations were then calculated using a standard curve with Curve Expert 1.4.

The number and area of skin lesions were recorded for each fish. In order to determine the lesion area, a clear template with a 0.25 $cm^2$ grid was placed over each lesion and the smallest possible area was recorded. All lesion areas per fish were then calculated. Sex, total body length, and weight were recorded. Each sampled fish was also implanted with a microchip (Mini Star ID, 1.4 × 8 mm) at an identical position, subcutaneously, close to the dorsal fin, in order to distinguish 'experimental fish' from accompanying fish, which were not sampled, later on. The latter were kept among the experimental fish to achieve the respective stocking densities. After sampling, the anesthetized fish were transferred to an aerated recovery tank and returned to their respective tanks after reflexes and reactivity were regained.

The final sampling was preponed by one week (in week 23 instead of week 24), because the fish had already reached their slaughter weight. Afterwards, all fish were weighed (in groups) and double checked for microchips. All fish with microchips were separated from the accompanying fish, properly stunned, killed, and separately disposed.

### 2.4. External Stressor

During the course of the experiment, at about the time of the third sampling, renovation work began in the adjacent building of the RAS facility used. This resulted in an unexpected stressor for the fish due to considerable noise impact. The demolition noise

occurred for several hours each day and continued for several weeks, especially covering samplings 3 and 4. Therefore, the experiment can be divided into three different phases, the regular experiment between sampling 1 and 2, the time where the demolition noise took place (samplings 3–4), and a phase without any further disturbance (sampling 5). The noise level was not acoustically measured, but as it involved chiseling off ceilings and rebuilding exterior facades with heavy machinery, it can be assumed that > 90 dB was reached [33].

*2.5. RNA Isolation and Multiplex Quantitative PCR*

Total RNA was extracted from individual spleens (*n* = 3 per group: stressed or unstressed fish under semi-intensive or super-intensive stocking densities from sampling 3 or sampling 5) using 1 mL TRIzol Reagent (Thermo Fisher Scientific, Waltham, MA, USA). After RNA purification using the RNeasy Mini Kit (Qiagen, Venlo, The Netherlands), the RNase-free DNase I (Qiagen) was used to digest residual DNA. Then, the concentration and the purity of the extracted RNA were assessed using a NanoDrop OneC spectrophotometer (NanoDrop Technologies, Wilmington, NC, USA).

The expression of four reference genes (*rna18s*, *rpl*, *actb*, *gapdh*) and twenty-two target genes (Table A1; [34]) was profiled in the extracted RNA specimens from all of the groups using the integrated fluidic circuit (IFC) technology of the Standard BioTools Gene Expression biochips. The multiplex quantitative PCR (qPCR) analyses were performed on one 48.48 IFC chip (Standard BioTools, South San Francisco, CA, USA) using the BioMark HD system (Standard BioTools). To this end, the total RNA was adjusted at a concentration of 10 ng $\mu L^{-1}$ and reverse-transcribed (42 °C, 30 min) using the Reverse Transcription Master Mix (Standard BioTools). The resulting cDNA aliquots were mixed with primers (100 $\mu$M) and the PreAmp master mix (Standard BioTools), and preamplified in 15 cycles (95 °C, 15 s; 60 °C, 4 min) in a TAdvanced thermocycler (Biometra, Jena, Germany). After this pre-amplification step, exonuclease I (New England BioLabs, Ipswich, MA, USA) was added to degrade single-stranded oligonucleotide primers, followed by a 30-min incubation period at 37 °C. Then, 43 $\mu$L TE buffer (Sigma, St. Louis, MO, USA) was added per sample and each 50-$\mu$L-cDNA sample was diluted in SsoFast EvaGreen Supermix with Low ROX (Bio-Rad) and 20 × DNA Binding Dye Sample Loading Reagent (Standard BioTools) to produce the sample mixes. After priming the 48.48-IFC chip in the MX Controller (Standard BioTools), the primers and the sample mixes together with one no-template (water) control were transferred to the assay and sample inlets on the primed 48.48-IFC chip. Finally, multiplex qPCR was conducted following the manufacturer's thermal protocol 'GE Fast 48.48 PCR + Melt v2.pcl'.

*2.6. Statistics*

Statistical tests were conducted with the Statistical Package for the Social Sciences (SPSS, v. 25, IBM Corp., 2017, Armonk, NY, USA) statistical software package. First, the resulting data were tested for distribution. Then, the non-parametric Kruskal-Wallis test by ranks and post hoc multiple range tests were conducted; Tukey's-HSD test for variance homogeneity and Dunnett-T3 test for variance inhomogeneity. Significance values were adjusted for several tests by the Bonferroni correction. All tests were performed with a significance level of $p < 0.05$.

For plasma cortisol and glucose, significances were only indicated within one sampling between stressed and unstressed fish of one single treatment group, as well as between stressed or unstressed fish of different treatment groups. No distinction was made between stressed and unstressed specimens for weight, length, CI, lactate, lesion number, and lesion area. Significances were indicated in this regard between treatment groups of one single sampling, as well as between similar treatment groups of different samplings.

The qPCR data were analyzed using the Fluidigm RealTime PCR Analysis Software (v. 4.5.2, South San Francisco, CA, USA) and normalized against the geometric mean of three suitable normalizer genes (*rna18s*, *actb*, *gapdh*). Copy numbers were calculated on the basis of ideal standard curves assuming a primer efficiency of 100%. GraphPad

Prism software (v. 9.1.0, San Diego, CA, USA) was used for the statistical analysis of the normalized qPCR data. Significant differences ($p < 0.05$) between the different groups were assessed using two-way analysis of variance (ANOVA) followed by a Holm-Šídák's post hoc test to correct for multiple comparisons.

## 3. Results

### 3.1. Fish Growth Performance

The mean initial and final weights and lengths of African catfish ($n = 100$) from the different stocking densities are given in Table 2. Before stocking, the weights of the representative sampling ($n = 100$) were statistically the same ($p > 0.05$). However, the lengths of the 100 fish measured in the intensive stocking density were found to be significantly different from both other groups. After 23 weeks, the mean weights in the semi-intensive density were about 200 g higher than in the higher stocking densities and significantly different from the super-intensive density ($p < 0.05$). In one semi-intensive and two super-intensive tanks, a few fish were caught due to organ sampling. That is why the stocking density is slightly lower at the end in these respective tanks. A comparative growth performance, including the CI of the fish from the three stocking densities over the course of the experiment based on sampling after 6 weeks each ($n = 18$), is given in the Appendix (Figures A1–A3).

**Table 2.** Initial and final weights and total body lengths of African catfish (mean $\pm$ standard deviation, SD, $n = 100$), $p > 0.05$, superscripts indicate significant differences between the groups.

| | Intended Stocking Density | $n$ Fish Tank$^{-1}$ | Mean Weight Fish$^{-1}$ [g] $\pm$ SD | Mean Lengths Fish$^{-1}$ [cm] $\pm$ SD | *de facto* Density Tank$^{-1}$ [kg m$^{-3}$] |
|---|---|---|---|---|---|
| Before stocking | Semi-intensive (0.84 kg m$^{-3}$) | 88 / 88 / 88 | $12.5 \pm 1.9$ | $12.1 \pm 0.7$[a] | 0.87 / 0.87 / 0.87 |
| | Intensive (1.68 kg m$^{-3}$) | 176 / 176 / 176 | $12.4 \pm 1.9$ | $11.6 \pm 0.8$[b] | 1.72 / 1.72 / 1.72 |
| | Super-intensive (3.35 kg m$^{-3}$) | 352 / 352 / 352 | $12.2 \pm 1.6$ | $12.0 \pm 0.6$[a] | 3.41 / 3.41 / 3.41 |
| After 23 weeks | Semi-intensive (100 kg m$^{-3}$) | 55 / 73 / 73 | $1830.5 \pm 596.7$ [a] | $57.3 \pm 6.3$ | 89.15 * / 106.40 / 106.85 |
| | Intensive (200 kg m$^{-3}$) | 161 / 161 / 163 | $1664.8 \pm 588.5$ [a,b] | $55.6 \pm 7.8$ | 230.62 / 212.34 / 214.96 |
| | Super-Intensive (400 kg m$^{-3}$) | 284 / 313 / 309 | $1615.4 \pm 451.7$ [b] | $56.1 \pm 5.1$ | 364.52 * / 401.13 / 396.60 * |

* deviation due to organ samplings.

The FCR tended to increase in all three stocking densities. Between the second-last and final sampling (T4 and T5, Table 3), the intensive stocking density showed the lowest FCR and, therefore, the best feed conversion rate. Throughout the entire experiment, the super-intense stocking density showed the comparatively weakest FCR. The SGR decreased with increasing age or weight of the fish. Comparing the three stocking densities, the highest SGR values were recorded under an intensive stocking density between T0 and T1, T2 and T4, and between T4 and T5 (Table 3).

### 3.2. Evaluation of Fish Welfare

Mortality varied between the rearing tanks, ranging from 2.59% to 14.62% (Table 4). The intensive stocking density showed the lowest mortality in total, while the super-intensive stocking density showed the highest mortality on average ($p > 0.05$). No increased mortality was observed during the phase of demolition noise.

**Table 3.** The feed conversion ratio (FCR) and specific growth rate (SGR) of African catfish 1 week after stocking (T0–T1), after 6 weeks (T1–T2), after 12 weeks (T2–T3), after 18 weeks (T3–T4), and after 23 weeks, at the final weighing of all fish (T4–T5). Calculations are based on sub-samplings (*n* = 18 per group), except totals (right column).

| Stocking Density | | T0–T1 | T1–T2 | T2–T3 | T3–T4 | T4–T5 | T0–T5 (in Total) |
|---|---|---|---|---|---|---|---|
| Semi-intensive | FCR | 0.7 | 0.7 | 0.8 | 0.7 | 1.1 | 0.77 |
| (100 kg m$^{-3}$) | SGR [% d$^{-1}$] | 7.0 | 6.0 | 2.2 | 1.7 | 0.9 | 3.0 |
| Intensive | FCR | 0.6 | 0.7 | 0.7 | 0.9 | 0.8 | 0.75 |
| (200 kg m$^{-3}$) | SGR [% d$^{-1}$] | 7.8 | 5.8 | 2.4 | 1.4 | 1.2 | 3.0 |
| Super-intensive | FCR | 0.7 | 0.7 | 0.8 | 0.8 | 1.2 | 0.83 |
| (400 kg m$^{-3}$) | SGR [% d$^{-1}$] | 6.8 | 6.0 | 2.3 | 1.6 | 0.7 | 2.9 |

**Table 4.** Mortality in the three treatment groups with semi-intensive, intensive, and super-intensive stocking densities (*p* > 0.05).

| Stocking Density | Mortality Tank$^{-1}$ [%] | Mean [%] $\pm$ SD |
|---|---|---|
| Semi-intensive (100 kg m$^{-3}$) | 2.59<br>9.56<br>9.20 | 7.12 $\pm$ 3.92 |
| Intensive (200 kg m$^{-3}$) | 3.89<br>4.32<br>3.16 | 3.79 $\pm$ 0.59 |
| Super-Intensive (400 kg m$^{-3}$) | 14.62<br>7.71<br>5.40 | 9.24 $\pm$ 4.80 |

The mean plasma cortisol responses of samplings 1, 2, 4, and 5 were in a range of 2.7 to 24.1 ng mL$^{-1}$ (Figure 1). Significant differences between unstressed and stressed fish, but not between the different stocking densities, were found within the respective samplings. At the first sampling, the cortisol responses were identical between stressed and unstressed fish kept at the super-intensive stocking density. The highest cortisol levels during the experiment, with most values between 17.1 and 32.2 ng mL$^{-1}$, were recorded in the third sampling, meaning the beginning of the second phase of the experiment, when the demolition noise likely affected the fish. However, during the fourth sampling, cortisol levels had already returned to the normal range, although the demolition noise was still ongoing. All relevant significances within a single sampling between unstressed and stressed fish in a treatment group, as well as between unstressed or stressed fish in a treatment group between different samplings, are given in Table A2.

Blood glucose mainly ranged between 3.8 and 8.1 mmol L$^{-1}$. There were significant differences between unstressed and stressed fish within the sampling groups. However, there were no significant differences between different stocking densities within a single sampling. Notably, the mean glucose values mainly were higher at the third sampling, and the interquartile ranges and upper Whiskers were mostly wider compared to the majority of boxplots at other samplings (Figure A4).

The lactate values mainly ranged between 2 and 8 mmol L$^{-1}$ (Figure 2). There were neither significant differences between the different stocking densities within one sampling nor between the samplings. Nevertheless, in the second phase of the experiment (under the influence of the demolition noise), i.e., from the third to the fourth sampling, all of the lactate values noticeably decreased and increased again to approximately their initial level at the fifth sampling.

Skin lesions due to agonistic interactions occurred during the entire experiment (Figures 3, 4 and A5). At the first and third sampling, several skin lesions per individual were found under all stocking densities, whereas at the second and fourth sampling,

the number of lesions decreased in general. At the first sampling, the semi-intensively stocked juveniles had the highest number of lesions, while at the fifth sampling, the adult fish had more lesions at intensive and super-intensive stocking densities compared to the semi-intensive stocking density. In particular, during the third sampling, meaning the beginning of the second phase (with noise impact), the highest number and largest skin lesions were determined in total. Significant differences ($p > 0.05$) were not found.

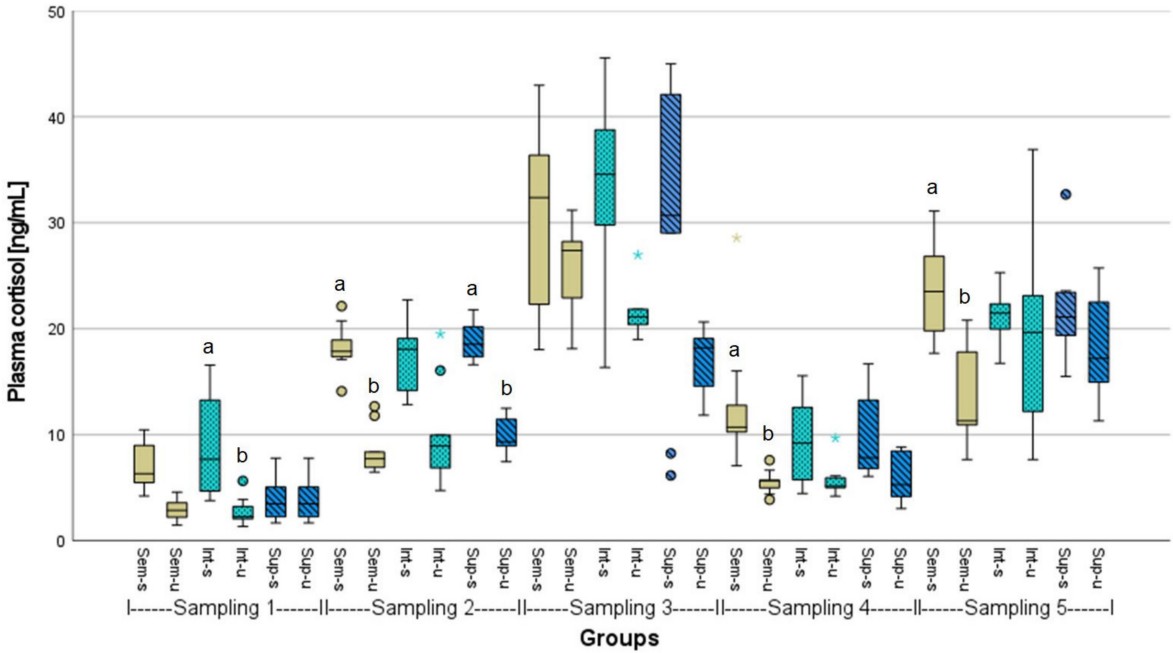

**Figure 1.** Plasma cortisol responses of stressed (s) or unstressed (u) African catfish (n = 9) under three different stocking densities (Sem = semi-intensive, Int = intensive, Sup = super-intensive). Kruskal-Wallis test with Tukey-HSD or Dunnett-T3 post hoc test, significances ($p < 0.05$) are only given within the respective samplings, marked by letters. Circlets = outliers; asterisks = extreme values.

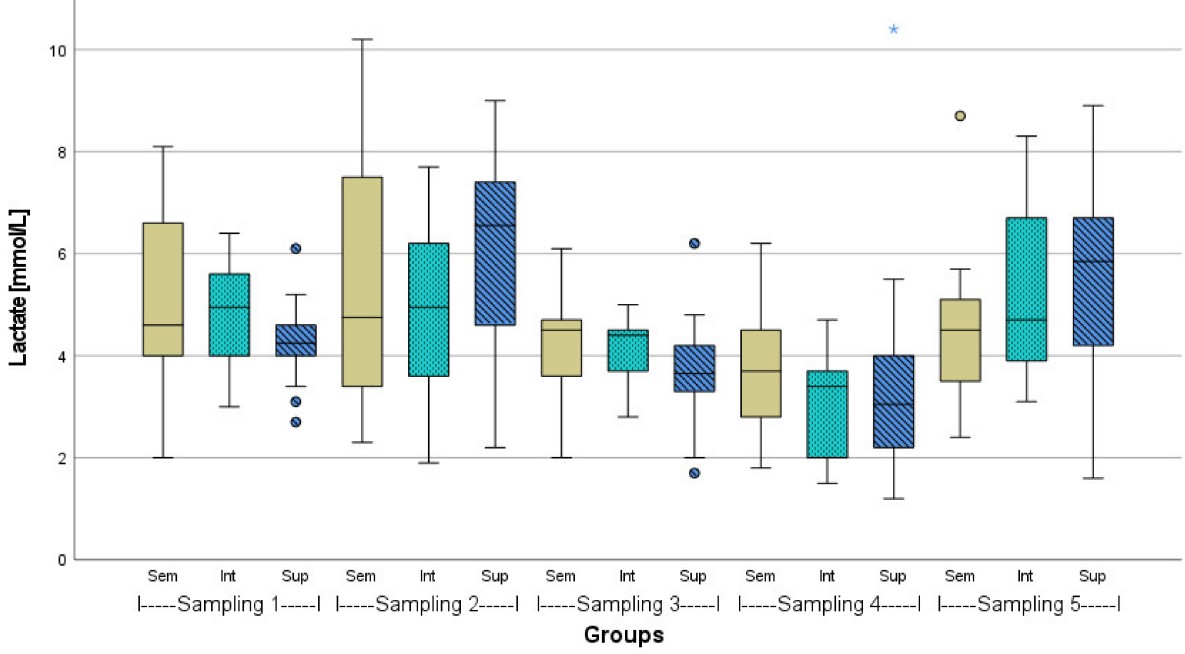

**Figure 2.** Lactate level of African catfish ($n = 18$) in three different stocking densities (Sem = semi-intensive, Int = intensive, Sup = super-intensive), $p > 0.05$. Circlets = outliers; asterisk = extreme value.

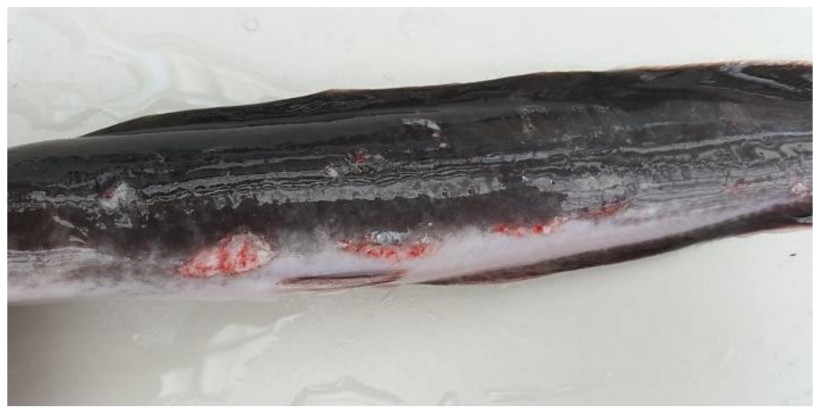

**Figure 3.** Skin lesions (bite marks) on an African catfish in the typical jaw form of this species as result of agonistic interactions. The photo was contrast-enhanced by 25%.

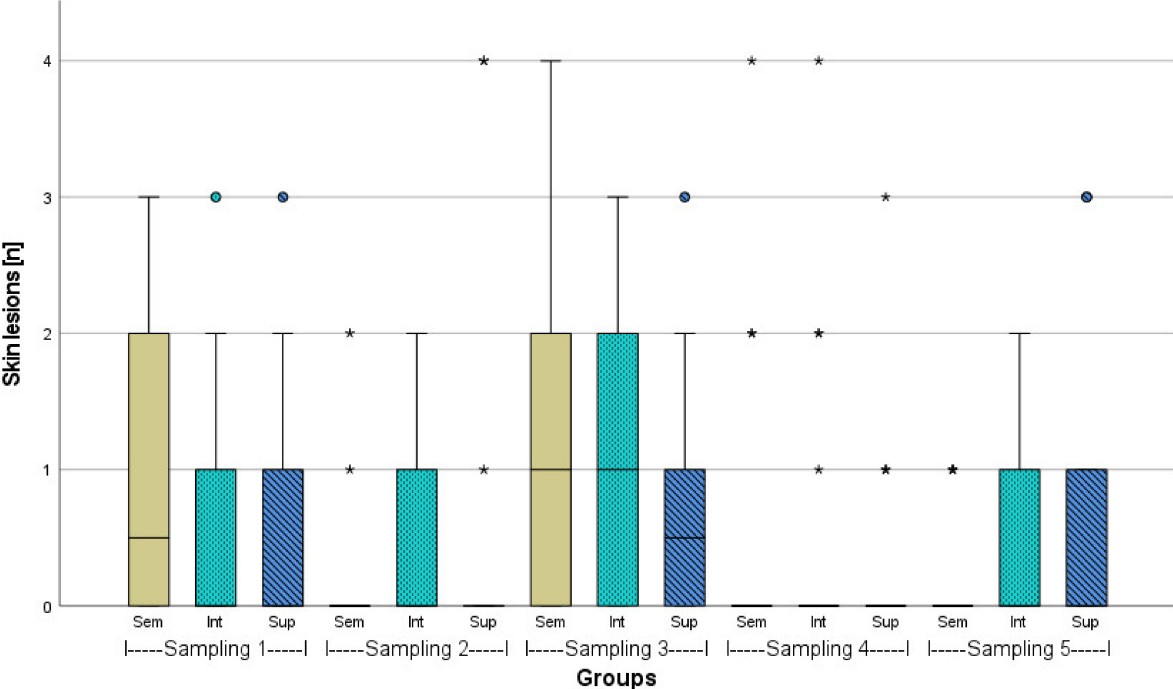

**Figure 4.** Number of external injuries (skin lesions) of African catfish (*n* = 18) in three different stocking densities (Sem = semi-intensive, Int = intensive, Sup = super-intensive), *p* > 0.05. In sampling 3, in Sem, a value of 6, and in Sup a value of 8 is not illustrated. Circlets = outliers; asterisks = extreme values.

Gene profiling focused on the third and fifth samplings, which were noticeable due to the high number of injured individuals. We hypothesized that skin lesions might entail not only stress, but also immune responses. Therefore, we quantified the transcript number of 22 selected genes in the spleen tissue of either unstressed or stressed African catfish kept under semi-intensive or super-intensive conditions (Figure 5). The spleen was chosen as the target tissue because it responds to both immune and stress-related stimuli. For instance, it has been well proven that the spleen stores red blood cells under stress conditions and/or during metabolic alterations.

The obtained qPCR data revealed a moderate impact on the level of the selected transcripts. At the third sampling, only *kmt2a* was slightly (1.2- to 1.3-fold, *p* < 0.002) downregulated in both unstressed and stressed African catfish kept under super-intensive versus semi-intensive conditions, and also in stressed versus unstressed individuals kept under both semi- and super-intensive conditions.

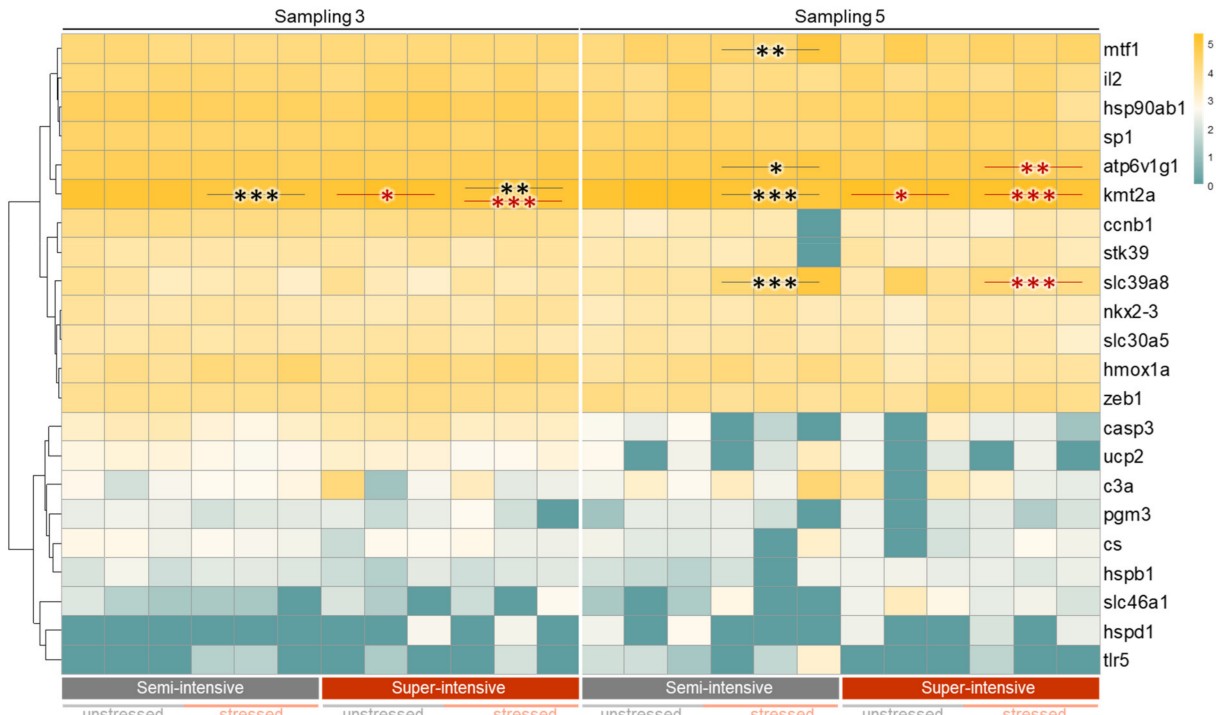

**Figure 5.** Hierarchical clustering of log10-transformed transcript numbers measured in the spleen of individual African catfish. The fish were kept under semi-intensive or super-intensive conditions, and individuals of both groups were exposed to stress, while a cohort remained unstressed (indicated below the heatmap). The sampling time points are indicated above the heatmap. The transcripts quantified are listed as gene symbols on the right margin; transcript levels are colored according to the scale on the left. Black asterisk(s) indicate a different expression in the stressed group compared to the matching unstressed group of the same stocking density and at the same sampling timepoint; red asterisk(s) indicate a different expression in the super-intensive group versus the semi-intensive group under the same conditions (either unstressed or stressed) and at the same sampling timepoint (*, $p < 0.05$; **, $p < 0.01$; *** $p < 0.001$).

At the fifth sampling, *kmt2a* was again differentially expressed between both unstressed and stressed African catfish kept under super-intensive versus semi-intensive conditions, and also in stressed versus unstressed individuals kept under semi-intensive conditions; however, in each case, the deviation was only between 1.1 and 1.5 (with $p < 0.03$). Similarly low (1.4- to 1.6-fold) but statistically significant ($p < 0.02$) were the deviations of the *atp6v1g1* levels between stressed versus unstressed African catfish kept under semi-intensive conditions and stressed individuals kept under super-intensive versus semi-intensive conditions.

In contrast, *slc39a8* was this study's most strongly regulated differentially expressed gene. In stressed African catfish kept under semi-intensive conditions, *slc39a8* was 9.7-fold down-regulated (with $p = 0.008$) compared to the unstressed cohort; in stressed individuals kept under super-intensive conditions, *slc39a8* was 3.9-fold up-regulated (with $p < 0.0001$) compared to the stressed catfish kept under semi-intensive conditions.

The levels of *mtf1* were moderately down-regulated by 2.2 (with $p = 0.008$) in stressed African catfish kept under semi-intensive conditions compared to unstressed fish. The transcript concentrations of the remaining 18 genes examined showed no significant alterations. Nevertheless, the 24-fold decrease in *casp3* transcripts (with $p > 0.05$) in stressed catfish kept under semi-intensive conditions compared to unstressed individuals may be worth mentioning.

## 4. Discussion

In the present study, the stocking density influenced the growth performance of African catfish under commercial rearing conditions. In addition, the effects of stocking density in relation to fish size, as well as external stress due to construction noise on different fish welfare indicators, were demonstrated.

In commercial RASs, African catfish can reach slaughter weight in about 6 months; in ponds, this may take longer [35]. The production time depends in particular on the water temperature and feed quality and quantity. In the present study, African catfish at all stocking densities reached appropriate slaughter weights after 23 weeks; 1 week earlier than expected. The water quality in the RAS was also found to provide suitable conditions for African catfish aquaculture without much potential to impair fish welfare. However, due to the very high stocking densities, this required an increased effort. Consequently, the cultivation conditions, as well as the growth performance, can be described as very good in each of the sampled groups.

Van de Nieuwegiessen et al. [28] indicated no growth difference between African catfish from different stocking densities ($96.9–485.6$ kg m$^{-3}$) after the final growth phase from approx. $1000–1500$ g fish$^{-1}$. In addition, partially contradictory findings have been reported. For instance, Toko et al. [36] described the influence of rather extensive stocking densities ($4–8$ fish m$^{-3}$) on juvenile African catfish (approx. $35–150$ g) in ponds, whereby growth performance significantly increased with increasing densities. However, other studies have reported lower growth with increasing densities in juveniles of approx. $30–100$ g in tanks at $35–125$ kg m$^{-3}$ [37–39], or in pond cages at max. $63$ kg m$^{-3}$ [40]. According to Hecht and Appelbaum [41], particularly in early developmental stages (larvae) of African catfish, stocking density can have an impact on growth. Li et al. [42] pointed out the heterogeneity of studies regarding the effect of stocking densities on growth.

In the present study, the super-intensively stocked fish showed the weakest growth performance, significantly different to the growth performance of semi-intensively stocked fish, which was approx. 13% better. This was also visible when comparing the FCR or CI (Table 3, Figure A3). Consequently, this study comes to similar conclusions regarding the influence of stocking density on growth as several previous studies; it is the case, even in commercial-scale RAS culture, that African catfish tend to grow slightly less under high stocking densities than under extensive conditions. This clearly contradicts the study by van de Nieuwegiessen et al. [28].

Mortality or survival of African catfish, with respect to stocking density, has again been differently described. Van de Nieuwegiessen et al. [22] reported a mortality rate of 2.5% under aquarium conditions, regardless of stocking density. Akinwole and Faturoti [43] described a survival rate of 75–93% (conversely, 7–25% mortality) for different developmental stages in commercial RASs. Palm et al. [31] found survival rates of 96% for extensively ($50$ kg m$^{-3}$) to 81% for intensively ($200$ kg m$^{-3}$) stocked fish. In the present study, mortality was not directly correlated with the increase in stocking density. The highest mortality was found in a super-intensive density tank, while it was overall lower in intensive and semi-intensive densities. However, African catfish survival rates can vary between different tanks, possibly due to differences in group composition, a factor that is difficult to control. Based on our experience, we see a higher risk of a greater percentage of African catfish dying under higher stocking densities. Unfortunately, in the present study it was not possible to investigate at what time how many fish died at each stocking density because the carcasses were quickly eaten by conspecifics.

Boerrigter et al. [44] described the physiological stress-related indicators, plasma cortisol, glucose, and lactate, prior to and post stress in juveniles of African catfish ($8–10$ g). The basal levels (unstressed) for cortisol were at $6.5–15.3$ ng mL$^{-1}$, for glucose at $3.7–5.4$ mmol L$^{-1}$, and for lactate at $1.3–2.2$ mmol L$^{-1}$, respectively. Directly, as well as 30 and 60 min after stressing, the fish plasma cortisol was at $12–48.5$ ng mL$^{-1}$, glucose at $4.2–6.8$ mmol L$^{-1}$, and lactate at $1.1–2.3$ mmol L$^{-1}$, respectively. Martins et al. [45] described, in particular after stress induction, higher levels of more than $100$ ng mL$^{-1}$

for plasma cortisol and up to 8 mmol $L^{-1}$ for glucose. Manuel et al. [46] depicted basal levels of cortisol for adults of African catfish (1–1.5 kg) at < 10 ng $mL^{-1}$; after handling, stress cortisol increased to 35 ng $mL^{-1}$, and after 3-h transportation to 50 ng $mL^{-1}$. These authors reported plasma glucose between approx. 3.5–6.5 mmol $L^{-1}$, with only minor elevations after stress induction. Van de Nieuwegiessen et al. [28] reported basal lactate levels in adult African catfish (1–1.5 kg) with approx. 3 mmol $L^{-1}$, and elevated levels of 3.6–4.9 mmol $L^{-1}$ after stress.

In the present study, the mean plasma cortisol and glucose levels of fish from each stocking density were in comparable ranges. Significant or at least trended differences with about two-fold elevated cortisol levels were frequently found between stressed and unstressed fish within one sampling. Both the basal levels and the elevations of cortisol due to induced stress were considered regular, according to the observations of Boerrigter et al. [44]; thus, the different stocking densities had at least no effect on this stress indicator. This is in contrast to the findings of van de Nieuwegiessen et al. [22], according to which an impaired cortisol response was observed under the lowest and highest stocking densities, and a down-regulation of ACTH or a depletion of the cortisol receptors due to chronic stress was suspected.

The overall glucose level can be considered relatively high in the measured ranges, but we assume that this could also be related to feeding, as earlier described by Polakof et al. [47]. In fact, feeding in our experiment was aimed at promoting excellent growth, which was achieved. In addition, we detected significant increases due to induced (acute) stress compared to basal levels, and no differences between semi-intensively and super-intensively stocked fish, suggesting feeding was the reason for an overall high glucose level. The mean lactate values were still within the normal range, albeit some fish had slightly elevated values compared to the abovementioned references. The induced stress was inappropriate to affect lactate levels because no physical activity or hypoxic stress was imposed by it. Lactate, as a gluconeogenic precursor, is directly related to blood glucose (also elevated) and generally increases with increased physical activity and/or oxygen deficiency. Both may have occurred under the high stocking densities tested.

The number of skin lesions on African catfish has been recorded in several studies [22,28,32,48–50]. The number largely varied between 0–5 per fish, depending on developmental stage and group composition. Studies addressing the area of skin lesions in African catfish are not known to the authors to date. In the present study, both the stocking densities and the respective developmental stage/fish size were relevant for the number and area of skin lesions. While early juveniles under semi-intensive stocking density had twice as many skin lesions on average as under intensive or super-intensive stocking density, it was reversed towards the end of the production period. Here, the matured fish under semi-intensive stocking densities showed very few skin lesions, while the number and area of bite wounds remained similar under intensive and super-intensive stocking densities. In any case, the occurrence of skin lesions may impair welfare. If possible, this should be considered for the different growth stages, also with regard to their current stocking density [51]. This behavioral change was already evident from other studies. According to Kaiser et al. [52] and van de Nieuwegiessen et al. [22], juvenile African catfish frequently show agonistic interactions, particularly under low stocking densities. This decreases under higher densities where the fish form "dense clusters [ . . . ] with constant movement and low aggression" [22] (p. 241). Information on adult African catfish is scarce, particularly at intensive or super-intensive stocking densities of commercial facilities. However, there is evidence that overall aggression decreases, resulting in fewer skin lesions [22,28].

Van de Niewegiessen et al. [22] described that skin problems and consequently infections might increase at stocking densities above 300 kg $m^{-3}$. Similar findings could not be generally confirmed in our study. However, a few fish in fact had to be removed from the super-intensive stocking density due to very severe and partially infected skin lesions.

The recording of somatic indices and the quantification of metabolic parameters of African catfish was complemented by gene-expression analysis to simultaneously quantify

22 genes associated with metabolism, immunity, and stress physiology. The transcript concentrations of most of these genes slightly changed or did not change significantly across the groups investigated, except for two genes. Both *slc39a8* and *mtf1* were significantly at least two-fold down-regulated in stressed African catfish kept under semi-intensive conditions compared to the unstressed fish. *Mtf1* encodes a transcription factor that is involved in cellular adaptation to a range of stress conditions [53]. *Slc39a8* codes for a transporter-like protein that maintains zinc homeostasis [54] and regulates cell migration [55]. It would be highly speculative to suggest possible functions of both gene products in African catfish, but the reduced levels of both *mtf1* and *slc39a8* might indicate an impending maladaptation of African catfish to adverse conditions. Previous single-gene expression analyses indicated few transcriptional parameters that could serve as potential indicator genes of stress responses in *Clarias* sp. [56,57]. The present study adds *mtf1* and *slc39a8* to the list of potential indicator genes that might be included in future studies on the welfare of African catfish.

Coppola et al. [58] described the noise as a physical stressor on animals leading to physiological, behavioral, and anatomical responses. It is well-known that anthropogenic noise affects many species [59]. Anthropogenic noise, which may include the background and partly loud noises of a mechanized aquaculture system, as well as other anthropogenic noise impacts (including construction and demolition noise), can negatively affect fish behavior, health, and welfare [60–62]. This presupposes that acoustic waves are transmitted from air into water. In the second phase of the experiment, particularly during the time of the third sampling, all treatment groups showed an increase in plasma cortisol and glucose levels, with noticeably larger interquartile ranges and upper Whiskers, slightly flattened lactate levels, and a larger number of skin lesions (including an increase in injured surface area), indicating that the fish were generally affected or stressed. Apparently, endogenous stress responses occurred more frequently in this second phase under noise exposure. The tentatively decreased lactate levels may indicate altered behavior, such as a reduction in swimming activity. At the same time, the African catfish reacted more aggressively, as evidenced by the increased incidence of skin lesions. Most likely, the demolition noise raised the stress levels in the African catfish and led to behavioral alterations. Therefore, it is recommended that external stressors in aquaculture, including loud noises, need to be avoided, especially if they persist for an extended period, preventing a chronic stress situation for the fish. It could have negative consequences for their wellbeing, health, and growth.

## 5. Conclusions

The growth of African catfish was very high at all stocking densities. However, at the end of the experiment, the semi-intensively stocked fish had significantly higher weights (by about 13%) than the super-intensively stocked fish. This study further demonstrates that particularly for early juvenile life stages of African catfish, intensive or super-intensive stocking density, whereas for matured fish, semi-intensive stocking density results in less aggression and fewer skin lesions. In addition, the fish welfare investigation revealed evidence of the influence of the demolition noise that occurred. We therefore suggest an adaptation of stocking density during the growth phases of African catfish after size-grading and the avoidance of external stressors in aquaculture to enhance production efficiency and fish welfare.

**Author Contributions:** Conceptualization, B.B. and L.C.W.; methodology, B.B.; validation, B.B.; formal analysis, B.B., A.R. and M.-C.H.; investigation, B.B., L.H., A.R., L.C.W., M.-C.H. and M.V.; resources, B.B. and L.C.W.; data curation, B.B., L.H., A.R. and M.V.; writing—original draft preparation, B.B.; writing—review and editing, B.B., A.R., L.C.W., M.-C.H., M.V. and H.W.P.; visualization, B.B. and A.R.; supervision, H.W.P.; project administration, B.B.; funding acquisition, H.W.P. All authors have read and agreed to the published version of the manuscript.

**Funding:** This research was funded by the project "Performance and process water management in commercial (integrated) aquaculture systems with African catfish (*Clarias gariepinus*) in Mecklenburg-Western Pomerania" (MV-II.1-LM-007: 139030000103, EMFF: 7302).

**Institutional Review Board Statement:** The animal study protocol was approved by the relevant Ethics Committee of the "Landesamt für Landwirtschaft, Lebensmittelsicherheit und Fischerei Mecklenburg-Vorpommern" (LALLF MV, approval number: 7221.3-1-042/20, date of approval: August 20, 2020).

**Data Availability Statement:** The data presented in this study are available on a reasonable request from the corresponding author. The data are not publicly available.

**Acknowledgments:** We thank Ulrich Knaus for project coordination and supporting with materials and feeds. We thank Philipp Sandmann, Ronja Siebert, Julian Bertram, and Tim Fuhrmann for their support with system maintenance and assistance during samplings (University of Rostock). We thank Ingrid Hennings, Brigitte Schöpel, and Franziska Witt for their excellent technical assistance (FBN, Dummerstorf).

**Conflicts of Interest:** The authors declare no conflict of interest. The funders had no role in the design of the study; in the collection, analyses, or interpretation of data; in the writing of the manuscript; or in the decision to publish the results.

## Appendix A

**Table A1.** Oligonucleotide-primer sequences derived from Clarias gariepinus, Ictalurus punctatus, or Pangasianodon hypophthalmus.

| Gene Symbol | Gene Product | Function | Sense Primer (5′→3′), Antisense Primer (5′→3′) | Source (Species; Accession Code) | Amplicon Length (bp) |
|---|---|---|---|---|---|
| **Reference genes:** | | | | | |
| *rna18s* | 18S ribosomal RNA | Structure of eukaryotic ribosomes | CTCTGCTGGACGATGGCTTAC, TCGATGAAGAACGCAGCCAGC | *C. gariepinus*; GQ465239 | 94 |
| *actb* | Actin-beta | Cell structure and motility, intercellular signalling | ACCACCACAGCCGAGAGAGAA, CTTCCAGCCATCTTTCCTTGGT | *C. gariepinus*; EU527191 | 204 |
| *gapdh* | Glyceraldehyde-3-phosphate dehydrogenase | Carbohydrate metabolism | TATGAAGCCCGCTGAGATCCC, GCCTCTTCTCACTTGCAGGGT | *C. gariepinus*; AF323693 | 106 |
| *rpl* | Ribosomal protein, large subunit | Structure of eukaryotic ribosomes | ACTAAATAGCAACTGATCCCTATC, GAATATCTGACCACTAAGATCCG | *C. gariepinus*; MW080924 | 134 |
| **Target genes:** | | | | | |
| *atp6v1g1* | ATPase H+ transporting v1 subunit g1 | Intercellular Fe homeostasis | CGGAAAAACCGCCGCTTGAAG, GACCAAGGAAGCCGCGGCAC | *P. hypophthalmus*; XM_026922532 | 106 |
| *c3a* | Complement component 3, variant a | Bacteria opsonisation and destruction | ATGTCTTTCGATGTCACGGTTTAT, TCGAACCAAGAGTAACGGCATG | *I. punctatus*; XM_017457024 | 114 |
| *casp3* | Caspase 3 | Apoptosis | CTCTTTATCATTCAGGCTTGTCG, GTACTCTACTGCTCCAGGTTATT | *I. punctatus*; XM_017473312 | 139 |
| *ccnb1* | Cyclin b1 | Control of the G2/M transition phase of the cell cycle | TCAAAAATCGGAGAGGTTACAGC, TGCACTTTGCTCCCTCTCTGG | *I. punctatus*; NC_030443 | 103 |
| *cs* | Citrate synthase | Aerobic metabolism | GGTGGTGAAGTGTCCGATGAAA, GCTATGGGCATGCTGTCCTGA | *I. punctatus*; XM_017487510 | 94 |
| *hmox1a* | Heme oxygenase 1 | Cellular response to xenobiotic stimulus | GATTCTTCTGTGTTCCCTGTATG, CCATCTACTTCCCTCAGGAGC | *I. punctatus*; XM_017491622 | 104 |
| *hsp90ab1* | Heat-shock protein 90 alpha family class b member 1 | Chaperone function, stress response | GAACATCAAGCTGGGCATCCAT, TTACTACATCACTGGTGAGAGCA | *I. punctatus*; XM_017456214 | 167 |
| *hspb1* | Heat-shock protein family b (small) member 1 | Differentiation of cell types, stress response | ACAGGACAACTGGAAGGTGAAC, GATTATCGGAAACCATGAGGAGA | *Clarias batrachus*; KT359728 | 107 |
| *hspd1* | Heat shock protein family d (hsp60) member 1 | Chaperone function, stress response | GCACGCTTGTCCTCAACAGGTT, AGACATGGCGATTGCTACTGGA | *I. punctatus*; XM_017469365 | 113 |
| *il2* | Interleukin-2 | Activation and proliferation of lymphocytes | GTCGGCCTGGGAAAAAGCCAAT, TTATGTGTTTGCACCAGACAACG | *I. punctatus*; XM_017474923 | 162 |
| *kmt2a* | Lysine-specific methyltransferase 2a | Regulation of early development and hematopoiesis | ATTGGGTCGAAATCGTGCTGTAT, ATGATAAGTCTTCAGTGGCAGGT | *I. punctatus*; XM_017490460 | 121 |
| *mtf1* | Metal regulatory transcription factor 1 | Catabolic regulation of cartilages | GTAGGAGGGCATTCAGGGAAC, AGTCAGAACGCTGCCCCCTC | *I. punctatus*; XM_017475296 | 146 |
| *nkx2-3* | NK2 homeobox 3 | Cell differentiation | TACAGGACAACCTGGTGGAAAG, ACAACTCTTGGTTTCCTGCTCTT | *I. punctatus*; XM_017464595 | 119 |
| *pgm3* | Phosphoglucomutase 3 | Carbohydrate metabolism | GACACAGGCAGGGCTGAATCT, CTTCGTACAGCACACTGTAACC | *I. punctatus*; XM_017494096 | 112 |
| *slc30a5* | Solute carrier family 30, member 5 | Zinc transportation | AATAGTCACCAAAAGACAGTGGAT, CATCGTTGTGCTCGAACAACAG | *I. punctatus*; XM_017459891 | 134 |
| *slc39a8* | Solute carrier family 39, member 8 alias ZIP8 | Cellular zinc uptake, protection from inflammation-related injury and death | TTTAACCTGATCTCAGCCATGTC, TATGTTCCCTGAGATGAATGCCA | *I. punctatus*; XM_017489708 | 151 |

**Table A1.** *Cont.*

| Gene Symbol | Gene Product | Function | Sense Primer (5′→3′), Antisense Primer (5′→3′) | Source (Species; Accession Code) | Amplicon Length (bp) |
|---|---|---|---|---|---|
| *slc46a1* | Solute carrier family 46, member 1 | Folate transportation | AATGGCGACATGCACAAGGGTAT, AGAACAGCCTTGCCCCAGGG | *I. punctatus*; XM_017491375 | 129 |
| *sp1* | SP1 transcription factor | Cell growth, apoptosis, differentiation and immune responses | AGCACAGCAGGTGATCAGGGA, GAGAAGCGTGCACATGTCCATA | *I. punctatus*; XM_017450095 | 119 |
| *stk39* | Serine/threonine kinase 39 | Stress response | TGTAGTTGTTGCTGCTAACCTTC, AGATCCCTGACGAGGTGAAGC | *I. punctatus*; XM_017469076 | 116 |
| *tlr5* | Toll-like receptor 5 | Detection of bacteria | GGCAGCATGGGAAAGGGAGTT, GTTAAGGCTCTGGATCTGTCCA | *I. punctatus*; NM_001200229 | 103 |
| *ucp2* | Uncoupling protein 2 | Regulation of production of reactive oxygen species, function of mitochondria | GGCTCCAGATCCAAGGGGAGA, CCACGTAGTCTCTACAACGGG | *I. punctatus*; XM_017489367 | 131 |
| *zeb1* | Zinc finger e-box binding homeobox 1 | Repression of interleukin-2 function | GCAGAGACCAGCGGCATGTAA, ATACGAGTGCCCCAACTGTAAAA | *I. punctatus*; XM_017483097 | 156 |

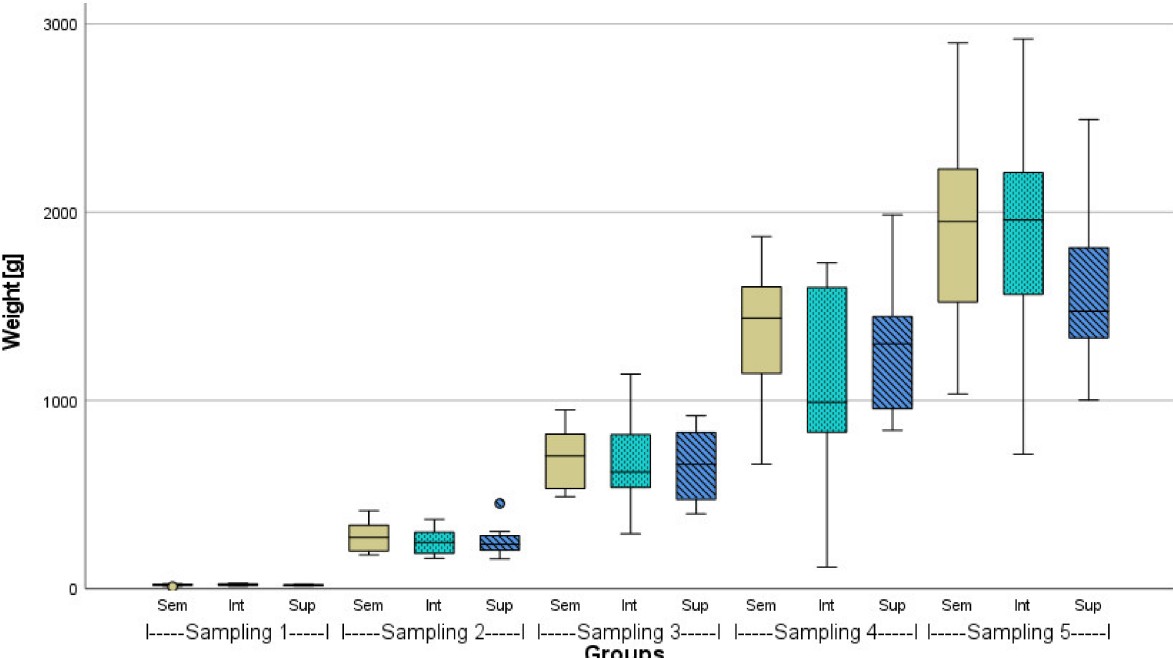

**Figure A1.** Weight gain of African catfish from three different stocking densities (Sem = semi-intensive, Int = intensive, Sup = super-intensive) over 23 weeks, $n = 18$, $p > 0.05$ in respective samplings. Circlet = outlier.

**Table A2.** Cortisol: Relevant significances within a single sampling between unstressed and stressed fish in a treatment group, and between unstressed or stressed fish in a treatment group between different samplings. Sem = semi-intensive; Int = intensive; Sup = super-intensive; the number before the stocking density abbreviation indicates the respective sampling; u = unstressed, s = stressed.

| Group Comparison | *p*-Value | Group Comparison | *p*-Value | Group Comparison | *p*-Value |
|---|---|---|---|---|---|
| **Within a sampling** | | **Within a sampling** | | **Within a sampling** | |
| 2Sem-u-2Sem-s | 0.022 | 1Int-u-1Int-s | 0.047 | 2Sup-u-2Sup-s | 0.038 |
| 4Sem-u-4Sem-s | 0.043 | **Between samplings** | | **Between samplings** | |
| 5Sem-u-5Sem-s | 0.032 | 1Int-u-2Int-u | 0.021 | 1Sup-u-2Sup-u | 0.040 |
| **Between samplings** | | 1Int-s-2Int-s | 0.026 | 1Sup-s-2Sup-s | 0.000 |
| 1Sem-u-2Sem-u | 0.042 | 1Int-u-3Int-u | 0.000 | 1Sup-u-3Sup-u | 0.000 |
| 1Sem-s-2Sem-s | 0.006 | 1Int-s-3Int-s | 0.000 | 1Sup-s-3Sup-s | 0.000 |
| 1Sem-u-3Sem-u | 0.000 | 1Inti-u-5Int-u | 0.000 | 1Sup-s-4Sup-s | 0.049 |
| 1Sem-s-3Sem-s | 0.000 | 2Int-u-3Int-u | 0.003 | 1Sup-u-5Sup-u | 0.000 |
| 1Sem-u-5Sem-u | 0.001 | 2Int-u-5Int-u | 0.043 | 1Sup-s-5Sup-s | 0.000 |
| 1Sem-s-5Sem-s | 0.000 | 4Int-s-2Int-s | 0.048 | 4Sup-s-2Sup-s | 0.031 |

**Table A2.** *Cont.*

| Group Comparison | *p*-Value | Group Comparison | *p*-Value | Group Comparison | *p*-Value |
|---|---|---|---|---|---|
| 2Sem-u-3Sem-u | 0.000 | 4Int-u-3Int-u | 0.000 | 4Sup-u-3Sup-u | 0.002 |
| 4Sem-u-3Sem-u | 0.000 | 4Int-s-3Int-s | 0.000 | 4Sup-s-3Sup-s | 0.002 |
| 4Sem-s-3Sem-s | 0.004 | 4Int-u-5Int-u | 0.001 | 4Sup-u-5Sup-u | 0.001 |
| 4Sem-u-5Sem-u | 0.018 | 4Int-s-5Int-s | 0.004 | 4Sup-s-5Sup-s | 0.008 |
| 4Sem-s-5Sem-s | 0.013 | | | | |
| 5Sem-u-3Sem-u | 0.021 | | | | |

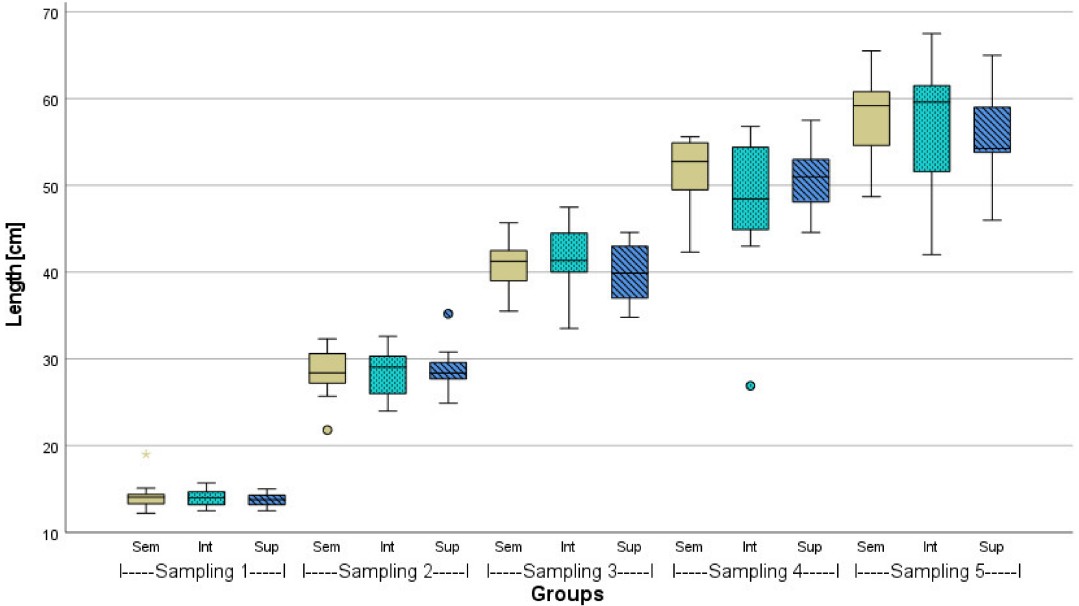

**Figure A2.** Total body length of African catfish from three different stocking densities (Sem = semi-intensive, Int = intensive, Sup = super-intensive) over 23 weeks, *n* = 18, *p* > 0.05 in respective samplings. Circlets = outliers, asterisk = extreme value.

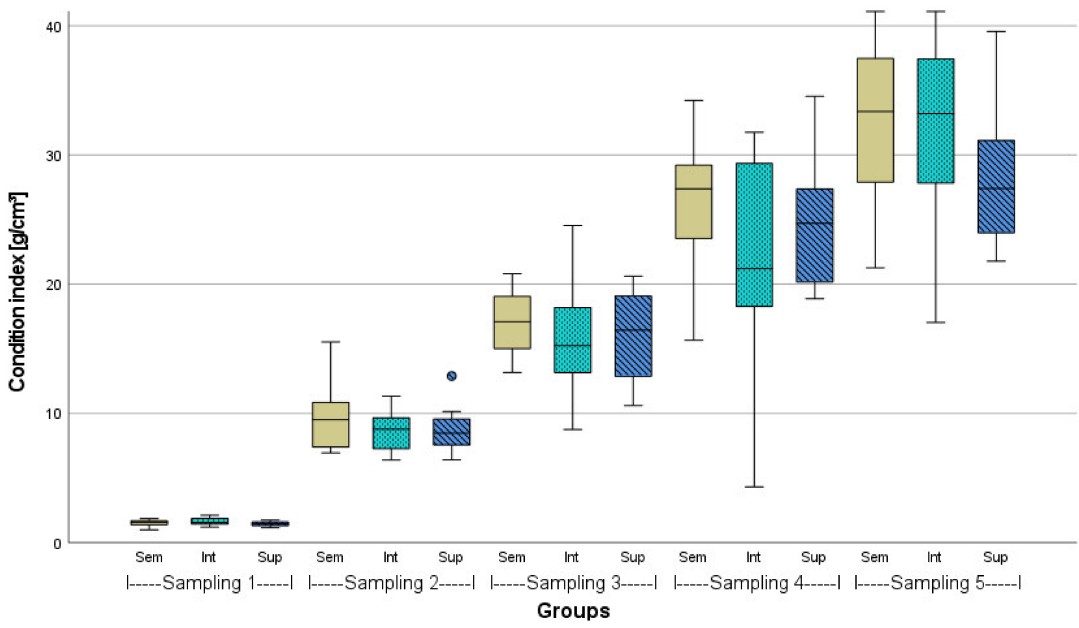

**Figure A3.** Condition indices of African catfish from three different stocking densities (Sem = semi-intensive, Int = intensive, Sup = super-intensive) over 23 weeks, *n* = 18, *p* > 0.05 in respective samplings. Circlet = outlier.

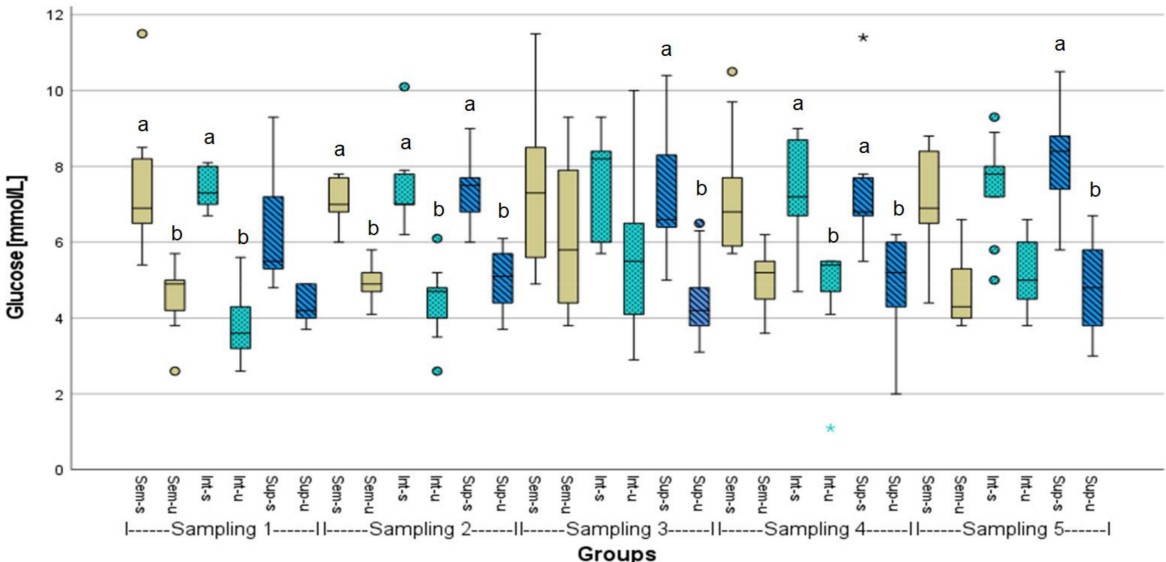

**Figure A4.** Glucose levels of stressed (s) or unstressed (u) African catfish (*n* = 9) under three different stocking densities (Sem = semi-intensive, Int = intensive, Sup = super-intensive). Kruskal-Wallis test with Tukey-HSD or Dunnett-T3 post hoc test, significances (*p* < 0.05) are only given within the respective samplings, marked by letters. Circlets = outliers; asterisks = extreme values.

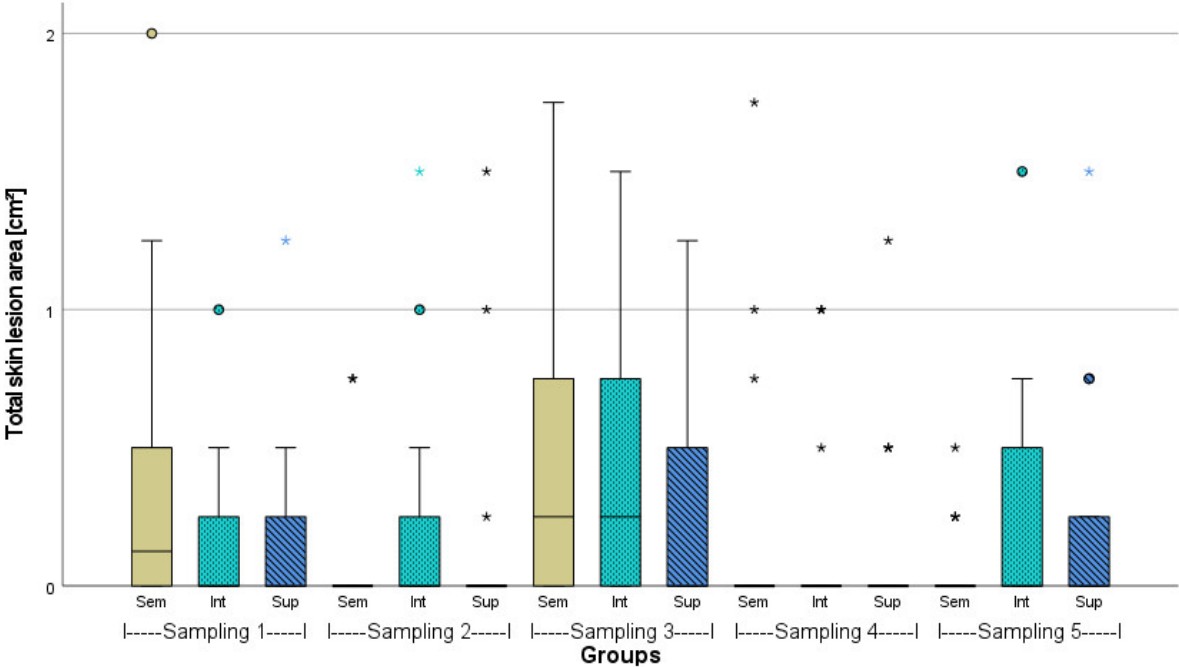

**Figure A5.** Total skin lesion area of African catfish in three different stocking densities (Sem = semi-intensive, Int = intensive, Sup = super-intensive), *n* = 18, *p* > 0.05. In sampling 3, in Sem, a value of 3, and in Sup a value of 4 is not illustrated. Circlets = outliers; asterisks = extreme values.

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
