# Peer review of "Effects of Stocking Density, Size, and External Stress on Growth and Welfare of African Catfish (Clarias gariepinus Burchell, 1822) in a Commercial RAS"

_fishes, doi:10.3390/fishes8020074_

Round 1
Reviewer 1 Report
my comments are in the file

Reviewer 2 Report
Authors have conducted the experiment and presented the results systematically. Unexpected stress factor of sound due to renovation should have been quantified scientifically like noise units in air and water at different depths, wave action etc. This is very relevant for quantitative estimation of noise during mechanical aquaculture systems like aeration, filtration, water pumping etc.,
All the stress factors like cortisol, lactate and skin lesions are not responded uniformly. Totally attributing skin lesions to stocking density may not be appropriate and it should have been correlated with microbial quality of the rearing system.
